# Comparison of Mean Values and Entropy in Accelerometry Time Series from Two Microtechnology Sensors Recorded at 100 vs. 1000 Hz During Cumulative Tackles in Young Elite Rugby League Players

**DOI:** 10.3390/s24247910

**Published:** 2024-12-11

**Authors:** Bruno Fernández-Valdés, Ben Jones, Sharief Hendricks, Dan Weaving, Carlos Ramirez-Lopez, Sarah Whitehead, Víctor Toro-Román, Michela Trabucchi, Gerard Moras

**Affiliations:** 1Research Group in Technology Applied to High Performance and Health, TecnoCampus, Department of Health Sciences, Universitat Pompeu Fabra, 08302 Barcelona, Spain; vtoro@tecnocampus.cat; 2Carnegie Applied Rugby Research (CARR) Centre, Carnegie School of Sport, Leeds Beckett University, Leeds LS1 3HE, UK; b.jones@leedsbeckett.ac.uk (B.J.); sharief.hendricks01@gmail.com (S.H.); d.a.weaving@leedsbeckett.ac.uk (D.W.); c.ramirez@leedsbeckett.ac.uk (C.R.-L.); s.whitehead@leedsbeckett.ac.uk (S.W.); 3Division of Physiological Sciences, Department of Human Biology, Faculty of Health Sciences, University of Cape Town, Cape Town 7925, South Africa; 4School of Behavioural and Health Sciences, Faculty of Health Sciences, Australian Catholic University, Brisbane, QLD 4014, Australia; 5England Performance Unit, Rugby Football League, Manchester M11 3FF, UK; 6Premiership Rugby, London SW1V 1PX, UK; 7Applied Sports Science and Exercise Testing Laboratory, The University of Newcastle, Ourimbah, NSW 2258, Australia; 8Department of Physical Activity and Sport, Faculty of Arts and Sciences, Edge Hill University, Ormskirk L39 4QP, UK; 9Leeds Rhinos Netball, Leeds LS6 3BR, UK; 10Department of Civil and Environmental Engineering, Universitat Politècnica de Catalunya (UPC), 08304 Barcelona, Spain; michela.trabucchi@upc.edu; 11National Institute of Physical Education of Catalonia (INEFC), 08038 Barcelona, Spain

**Keywords:** frequency, sport technology, rugby, tackle

## Abstract

Several microtechnology devices quantify the external load of team sports using Global Positioning Systems sampling at 5, 10, or 15 Hz. However, for short, explosive actions, such as collisions, these sample rates may be limiting. It is known that very high-frequency sampling is capable of capturing changes in actions over a short period of time. Therefore, the aim of this study was to compare the mean acceleration and entropy values obtained from 100 Hz and 1000 Hz tri-axial accelerometers in tackling actions performed by rugby players. A total of 11 elite adolescent male rugby league players (mean ± SD; age: 18.5 ± 0.5 years; height: 179.5 ± 5.0 cm; body mass: 88.3 ± 13.0 kg) participate in this study. Participants performed tackles (*n* = 200), which were recorded using two triaxial accelerometers sampling at 100 Hz and 1000 Hz, respectively. The devices were placed together inside the Lycra vests on the players’ backs. The mean acceleration, sample entropy (SampEn), and approximate entropy (ApEn) were analyzed. In mean acceleration, the 1000 Hz accelerometer obtained greater values (*p* < 0.05). However, SampEn and ApEn were greater with the 100 Hz accelerometer (*p* < 0.05). A large relationship was observed between the two devices in all the parameters analyzed (R^2^ > 0.5; *p* < 0.0001). Sampling frequency can affect the quality of the data collected, and a higher sampling frequency potentially allows for the collection of more accurate motion data. A frequency of 1000 Hz may be suitable for recording short and explosive actions.

## 1. Introduction

The use of microtechnology devices to quantify the external load in team sports has increased exponentially over the last decade [1,2,3,4,5]. However, most of the research about external load has been performed using Global Positioning System (GPS)-derived variables (e.g., distance, high-speed running, accelerations, and decelerations) [6,7]. Most of these devices sample at 5, 10, or 15 Hz, and all of these frequencies have been validated for measuring the movement demands of team sport [8,9,10,11]. Technological advances allow athletes performing indoor to be monitored using a Local Positioning System (LPS), which has an improved validity and reliability compared to GPS and can sample up to 20 Hz [12]. However, for short actions like collisions in sports such as rugby union, rugby league, or American Football, these sampling frequencies could be a limiting. It has been reported that GPS devices at 1 Hz may be unable to record movements that take <1 s [11]. The literature also reports that 10 Hz devices are able to measure the smallest change in acceleration and deceleration, while the 5 Hz units may not [8]. Furthermore, it appears that a higher sampling rate improves the reliability of GPS measurements [13].

Microtechnological devices are multicomponent and typically contain a GPS, accelerometer, gyroscope, and barometer [14]. Accelerometers may provide additional information about human movement [6,15,16,17]. Usually these accelerometers sample at a frequency of 100 Hz [6] and some even at 1000 Hz [18]. The physical demands of rugby league are characterized by short-duration, high-intensity intermittent drills with frequent physical collisions between players, known as tackles [19,20,21]. Understanding the physical demands of a tackle in real match situations is necessary for the design and development of training drills [1,5,22]. Therefore, there is a requirement to measure short and explosive actions like collisions [22]. Variability of athletes during the performance of a movement, such as a collision, must be investigated as a key element, to identify the amount of perturbation in a specific action [23,24]. Therefore, human movement variability may provide an additional tool for the quantification of collision demands in team sport.

The analysis of human movement has evolved to assess the variability of a measure, targeting the detection of changes in fluctuations and spatiotemporal characteristics of outcomes [25,26]. Within the past 20 years, entropy analysis has become popular as a measure of system complexity and used to describe changes in postural control [27] and to assess running [28] and tactical behavior in soccer [29]. It has also been validated for detecting changes in movement variability during resistance training in elite rugby union and soccer [30,31,32]. In these previous studies, the accelerometer sampling frequency was 1000 Hz, and entropy was calculated via the summation of vectors of total acceleration (AcelT) in three planes: mediolateral (x), anteroposterior (y), and vertical (z) components of the acceleration signal. Two of the most widely used and successful entropy estimators are approximate entropy (ApEn) [33] and sample entropy (SampEn) [34]. ApEn quantifies the similarity probability of patterns of lengths m and m + 1. SampEn is a similar statistic, measuring the probability of subsequences being close at two lengths, m and m + 1. However, SampEn does not include self-comparisons, though it exhibits greater consistency than ApEn [35].

Previous studies (analyzing actions that ranged in duration from 5 s to 10 min) have reported decreases in SampEn as the sampling frequency increased during walking [36] or during an isometric contraction [37]. These authors recommended sampling frequencies between 100 and 250 Hz, similar to those also reported by Marmelat et al. [38], who concluded that the use of sampling frequencies of 120 Hz and 240 Hz provides similar results, but that the use of 60 Hz may alter Detrended Fluctuation Analysis (DFA) values in gait kinematics. 

The authors of a previous study found that movement variability progressively decreases with cumulative tackle events, particularly among backs and in defensive roles. Entropy measures can be used by practitioners as an alternative tool to analyze the temporal structure of variability in tackle actions and to quantify the load of these actions by position [24]. Given the characteristics of the tackle, we believe that to analyze the variability of movement in this short, explosive action, an accelerometer sampling at a frequency of 100 Hz or above is necessary. Therefore, the aim of this study was to compare mean acceleration and entropy values obtained from 100 Hz and 1000 Hz like accelerometer units in tackle actions performed by young professional rugby league players.

## 2. Materials and Methods

### 2.1. Participants

Eleven elite adolescent male rugby league players participate in this study (mean ± SD, age; 18.5 ± 0.5 years, height; 179.5 ± 5.0 cm, body mass; 88.3 ± 13.0 kg; six backs and five forwards). All participants were selected from a single professional rugby league academy based in England. Prior to volunteering, the experimental protocol was explained to all participants both verbally and in writing, with a written statement of consent signed. The procedures complied with the Declaration of Helsinki (2013) and were approved by the Leeds Beckett University Research Ethics Committee (21/20118/CEICEGC). To participate in this study, the following criteria had to be met: (i) at least 5 years of rugby-playing experience; and (ii) no injury or illness at the time of this study or 1 month prior.

### 2.2. Procedures

Participants performed a drill encompassing one-on-one tackles, divided into tackling (i.e., tackling an opponent) and tackled (i.e., being tackled by an opponent while carrying a ball) events. The players started in front of each other, and when the coach marked the start, the players moved two meters in the opposite directions from each other and then changed directions to execute the tackle at the central point (Figure 1). The players were divided by positions (e.g., forwards or backs), so that they were always paired with a player of their same position. These drills were structured in four blocks, and each block consisted of six tackling and six tackled activities in a random order. The experimental protocol began with a standardized warm-up. Participants were instructed and encouraged to tackle with maximum effort. During tackling actions, participants alternated between shoulders (i.e., three tackles using the dominant shoulder and three tackles using the non-dominant shoulder) within each block. Ninety seconds of passive recovery was prescribed between each block. Professional coaches directed the sessions to ensure session safety and ecological validity.

### 2.3. Equipment

Two different brands of microtechnological devices were used: First, an Optimeye S5 (Catapult Innovations, Melbourne, Australia) with a built-in triaxial accelerometer capable of recording at a maximum of 100 Hz and measuring accelerations in gravitational forces (g) in three planes (x, y, z) of movement (anteroposterior, vertical, and mediolateral). Second, a WIMU device (RealtrackSystems, Almería, Spain) with a built-in triaxial accelerometer capable of recording at a maximum of 1000 Hz, as well as measuring accelerations in gravitational forces (g) in the three planes of movement.

Participants wore two microtechnological inertial measurement units (IMUs), one with an accelerometer with a sampling frequency of 100 Hz, and the other with an accelerometer with a sampling frequency of 1000 Hz. The devices were placed together inside a Lycra vest provided by the device manufacturer, selecting the most appropriate size for each athlete. Proper sizing for each athlete reduces artifacts from fabric movement and overestimation of peak acceleration [6,39]. The order of the devices within the vest was switched halfway through the protocol.

### 2.4. Data Analysis

The raw acceleration signal was extracted from each device (Figure 2 and Figure 3) and processed using a summation of vectors (AcelT) in three axes, mediolateral (x), anteroposterior (y), and vertical (z), calculated according to Moras et al. [30].

The (AcelT) signal was cut, separating each collision in each device, obtaining 200 signals for each device. Mean acceleration, approximate entropy (ApEn), and sample entropy (SampEn) for each signal were calculated. Entropy was calculated according to Goldberger et al. [40] and through dedicated functions programmed in MATLAB^®^ (MatLab 2021b, The MathWorks, Natick, MA, USA).

### 2.5. Statistical Analysis

A comparison between the two devices was conducted using Excel [41] to calculate the mean bias, typical error of estimation (TEE), and Pearson’s correlation, all at a 90% confidence interval. The standardized mean bias was interpreted as *trivial* (<0.19), *small* (0.2–0.59), *medium* (0.6–1.19), or *large* (1.2–1.99) [42]. The standardization of the typical error of estimation was interpreted as *trivial* (<0.1), *small* (0.1–0.29), *moderate* (0.3–0.59), or *large* (>0.59) [43]. The magnitude of the correlation was interpreted as *trivial* (<0.1), *small* (0.1–0.29), *moderate* (0.3–0.49), *large* (0.5–0.69), *very large* (0.7–0.89), or *nearly perfect* (0.9–0.99) [44].

A paired-samples t-test was also conducted for each of the three analyzed variables (mean acceleration, SampEn, and ApEn) using PASW Statistics 21 (SPSS, Inc., Chicago, IL, USA). Linear regression analysis was performed using R (v4.1.2, R Foundation for Statistical Computing, Vienna, Austria). The alpha was set as *p* < 0.05 for all analyses.

## 3. Results

Table 1 displays the mean values of acceleration, SampEn, and ApEn for the different analyzed devices.

In terms of mean acceleration, the 1000 Hz accelerometer yielded significantly higher results (*p* < 0.05). Meanwhile, both SampEn and ApEn entropies were higher with the 100 Hz accelerometer (*p* < 0.05). *Moderate* bias was observed for mean acceleration values when comparing data obtained at 100 Hz against those obtained at 1000 Hz, along with a *large* typical error of estimation and a *very large* correlation. In contrast, for entropy values, there was a *moderate* bias and a *large* typical error of estimation for both SampEn and ApEn, with a *very large* correlation for SampEn and a large one for ApEn.

In Figure 4, the regression line equations for mean acceleration, SampEn, and ApEn between the two microdevices analyzed are displayed. For all examined parameters, R² values greater than 0.5 were observed, with a *p*-value of < 0.0001, indicating *large* to *very large* significant correlations. Specifically, Figure 4A shows R^2^ = 0.72, Figure 4B shows R^2^ = 0.50, and Figure 4C shows R^2^ = 0.27.

## 4. Discussion

The aim of this study was to compare mean acceleration and entropy (SampEn and ApEn) between two accelerometers with different sampling frequencies (100 Hz vs. 1000 Hz) during tackling tasks performed by male adolescent rugby league players. The present research found that for short and intense actions like tackles, the 1000 Hz accelerometer yielded higher values in mean acceleration. However, SampEn and ApEn were higher with the 100 Hz accelerometer. According to the data, at a sampling frequency of 100 Hz, an acceleration signal may miss important information about mean acceleration, especially in peaks, and may result in very high entropy values, influenced by the sampling frequency in short and explosive actions like tackles. Despite this, the strong correlation between devices indicates the possibility of understanding inter- and intra-subject behavior in these exercises. This underscores the importance of not mixing data obtained at different sampling frequencies.

In Figure 2 and Figure 3, both signals may appear quite similar at first glance. However, there are differences in both mean acceleration values and entropy values. To better visualize these differences graphically, we represent the data points corresponding to all measurements taken and zoom in to capture two of the three main peaks produced during the movement (Figure 3). In this case, we can clearly see that the signal recorded at 100 Hz is much more chaotic and irregular, which explains the differences observed in entropy values, with higher values when recording at 100 Hz. Furthermore, in the zoomed-in image, we can also observe that the 100 Hz recording does not allow for proper signal reconstruction, especially affecting the recording of acceleration peaks at the moment of impact in tackles.

Studies investigating the influence of the sampling frequency have analyzed various frequencies, ranging from 25 Hz to 750 Hz. However, it is uncommon to use sampling frequencies of 1000 Hz or similar values [37]. Raffalt et al. [36] have published a study in which they compare entropy values obtained for the same exercise recorded simultaneously using different sampling frequencies. In this case, the signal to which entropy calculation was applied was a video signal recorded at 120, 240, and 480 Hz. More recently, Raffalt et al. [37] investigated the effect of different sampling frequencies (1000, 750, 500, 250, 100, 50, and 25 Hz) on SampEn calculated from torque data recorded during submaximal isometric contractions. The authors reported that SampEn significantly increased at sampling frequencies below 100 Hz and remained unchanged above 250 Hz. Although none of the previous studies were conducted with an accelerometer, the results are in line with those reported in the present study, where higher sampling frequencies resulted in lower entropy values and where differences exist between entropy values calculated from different sampling frequencies. This may be due to the lower density of data points in the time series per unit of time making entropy less predictable and consequently raising entropy values. Therefore, for the tackle action, it is likely ideal to record at a frequency between 250 and 1000 Hz.

When recording biological signals, it is essential to select an appropriate sampling frequency [45,46]. According to Nyquist’s theorem, the sampling frequency should be at least twice the size of the highest-frequency component in the signal of interest [47]. It is worth noting that sampling at too high a frequency when observing a phenomenon that evolves with low-frequency oscillations could lead to the collection of redundant information [41]. Similarly, reducing the sampling rate could affect the analysis of regularity and variability in the time domain [43]. When using continuous data, it is important to keep in mind that the nervous system does not have an infinite resolution. Reflexes and muscle activity modulations are controlled at a millisecond level [48]. Therefore, sampling data at a rate beyond 1000 Hz can lead to redundant information [41].

Despite the large significant relationships between both sampling frequencies for all the parameters analyzed, which suggest a relationship to the increase or decrease in entropy at the two different sampling frequencies, the high values of bias and estimation errors point to errors in absolute values. Therefore, we stress that it is crucial not to mix data obtained at different sampling frequencies for training load analyses. Additionally, the selection of sampling frequencies should be based on the spatiotemporal characteristics of the movement for each task/action and the duration of those tasks. This is especially important in short and intense exercises such as tackle actions. In these actions, not only do entropy values change but the mean acceleration values are also substantially altered, and 100 Hz sampling does not capture all movement peaks that could be relevant for load control. However, subtle nuances in movement changes during tackling can be detected, providing a more realistic and detailed reference of how athletes move during these types of technical actions, which demand high levels of physical conditioning.

This study is not without limitations. It only focuses on one of the numerous short and explosive actions found in team sports. Moreover, the tackling actions were performed in controlled one-on-one situations, which may differ from the coordinative and cognitive realities of tackling scenarios in actual gameplay. Furthermore, it compared two widely different sampling frequencies. Additionally, only two commercial brands were compared. It would be interesting for future studies to analyze various, more finely spaced sampling frequencies, collected with different sensors, and across different sporting movements and training exercises. Also, future studies should compare values obtained among sensors with sampling frequencies ranging from 100 to 1000 Hz to determine the optimal sampling frequency for microtechnological sensors in order to avoid losing information in short and explosive actions.

## 5. Conclusions

In conclusion, given the temporal characteristics of strength tasks and explosive actions in team sports, a sampling frequency of 100 Hz may be too low to accurately capture the recorded phenomena, especially for entropy analyses. Nonetheless, this sampling frequency maintains a good correlation regarding behavior over multiple repetitions. We stress that it is crucial not to mix data collected at different sampling frequencies for result comparisons.

## 6. Practical Applications

IMUs remain one of the most widely used tools in monitoring and load control in team sports. Improving our understanding of the efficacy of data collected depending on the sampling frequency can assist sensor manufacturers in sports applications and sport science professionals in making data-driven decisions that allow for continuous improvement. For instance, an improved understanding may lead manufacturers to develop sensors that allow users to quickly and easily select the desired sampling frequency.

According to the scientific literature [9,49], the sampling frequency can affect the quality of the data collected, with a higher sampling frequency potentially allowing for the collection of more accurate motion data. A frequency of 1000 Hz may thus be suitable for recording short and explosive actions. It is worth noting, however, that we consider it important to continue research in this area, to clarify the optimal sampling frequency based on the type of sports action being monitored. This will minimize errors in decision-making based on load metrics obtained from these sensors. A sampling frequency that is too low could lead to errors in interpreting load, movement variability, or the intensity of actions.

## Figures and Tables

**Figure 1 sensors-24-07910-f001:**
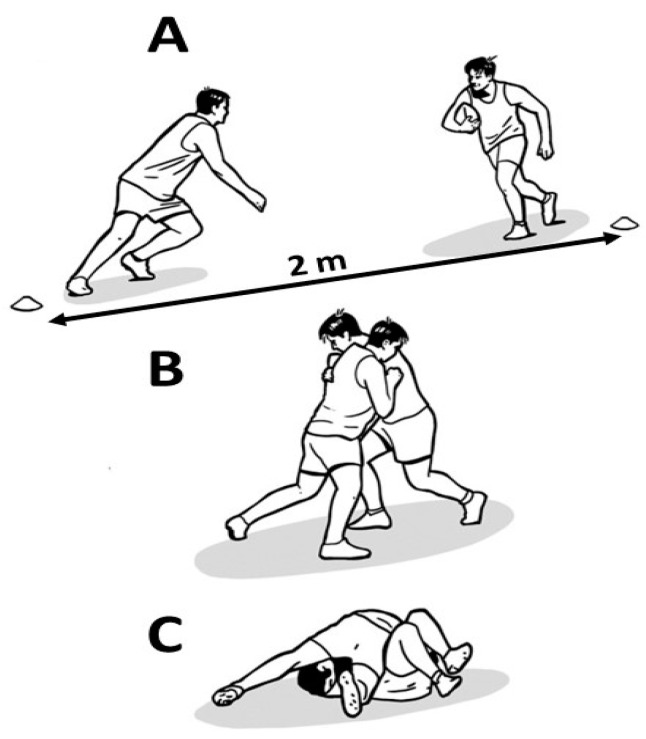
One-on-one tackle drill.

**Figure 2 sensors-24-07910-f002:**
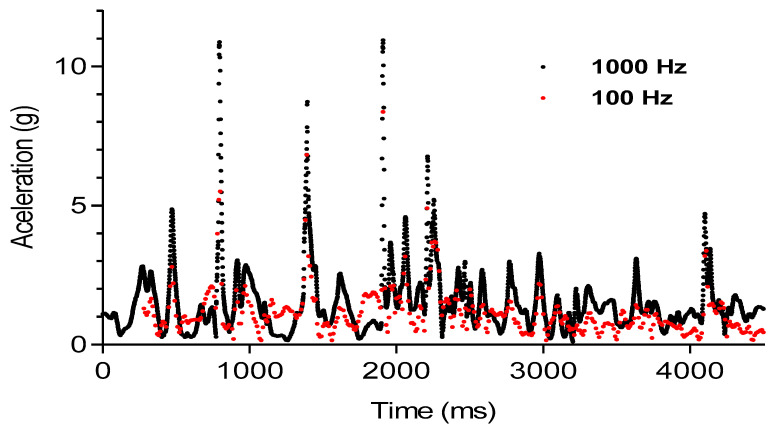
Raw signals from a randomly selected tackle out of the 200 signals obtained. Signal from both devices during a tackling action; black color: 1000 Hz device; red color: 100 Hz device. This shows the signals of the same exercise recorded with both accelerometers, one superimposed over the other.

**Figure 3 sensors-24-07910-f003:**
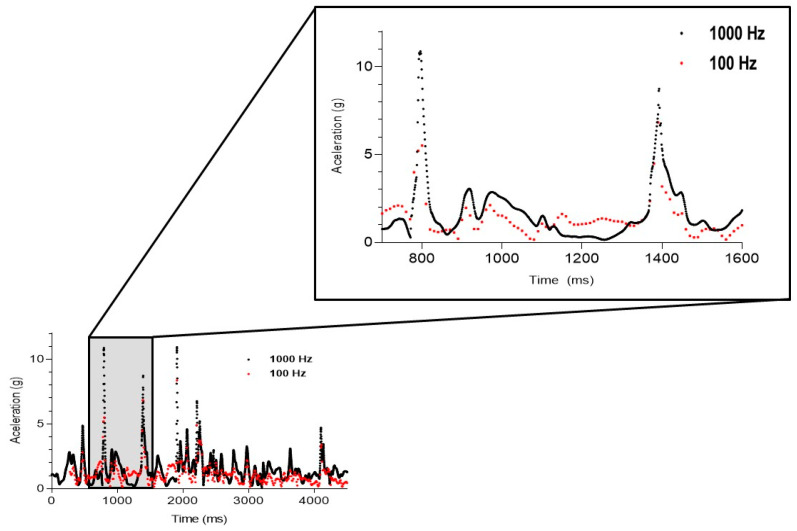
Enlarged figure of 2 movement peaks during the tackling action; black color: 1000 Hz device; red color: 100 Hz device.

**Figure 4 sensors-24-07910-f004:**
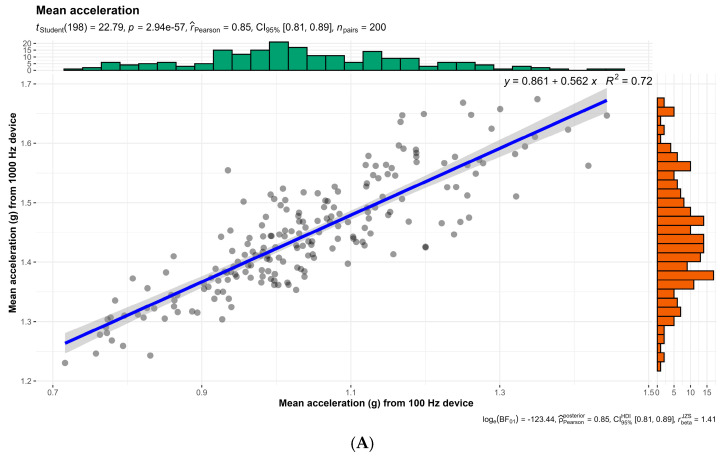
(**A**). Correlations and linear regression equations for mean acceleration between the two devices. CI: confidence interval. (**B**): Correlations and linear regression equations for SampEn between the two devices. CI: confidence interval; SampEn: sample entropy. (**C**): Correlations and linear regression equations for ApEn between the two devices. CI: confidence interval; ApEn: approximate entropy.

**Table 1 sensors-24-07910-t001:** Values of the mean acceleration, SampEn, and ApEn.

Parameter	Accelerometer 1000 Hz	Accelerometer 100 Hz	Bias (90% CI)	SEE (90% CI)	*r* (90% CI)
Mean acceleration (g)	1.44 ± 0.10	1.00 ± 0.10 *	0.85 (0.78–0.92) *moderate*	0.62 (0.54–0.71) *large*	0.85 (0.82–0.88) *very large*
SampEn (a.u.)	0.08 ± 0.02	0.56 ± 0.13 *	6.42 (5.93–7.00) *large*	1.00 (0.85–1.19) *large*	0.71 (0.64–0.76) *very large*
ApEn (a.u.)	0.15 ± 0.03	0.67 ± 0.09 *	2.91 (2.69–3.17) *large*	1.65 (1.34–2.11) *large*	0.52 (0.43–0.60) *very large*

SampEn: sample entropy; ApEn: approximate entropy; * *p* < 0.05 differences 1000 vs. 100; SEE: standard error of estimation; *r*: Pearson’s correlation; CI: confidence interval.

## Data Availability

The original contributions presented in this study are included in the article, and further inquiries can be directed to the corresponding author.

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
