# Peer review of "Comparison of Mean Values and Entropy in Accelerometry Time Series from Two Microtechnology Sensors Recorded at 100 vs. 1000 Hz During Cumulative Tackles in Young Elite Rugby League Players"

_sensors, 2024, doi:10.3390/s24247910_

Round 1
Reviewer 1 Report
Comments and Suggestions for Authors
The authors compared the accelerations and entropy values recorded from accelerometers sampling at different frequencies during tackling tasks performed by junior rugby league players. The results presented hold implications for practitioners assessing the training loads of players. However, there are a number of small issues that I would like the authors to address before the manuscript is accepted for publication.
Abstract, line 42: Change "was superior" to "were greater"
Introduction line 57: Change to "...compared to GPS and can sample up to 20 Hz."
Introduction, line 62: Change to "...the 5 Hz units may not [14]. It also appears..."
Introduction, line 65: Remove the parenthesis that appears after in the in-text citation.
Introduction, lines 73-74: It is not clear what you mean by "...must be perceived as a key element to identify the amount of perturbation in a specific action." You are presumably quantifying that the variability in training tasks can be used as a marker of perturbations and that such perturbations are important for skill acquisition, as per the ecological dynamics approach? Please rewrite this sentence to ensure that your intended meaning is clear.
Introduction, line 80: Move the period (full-stop) following the in-text citation.
Introduction, line 85: Change to "...vertical (z) components of the acceleration signal."
Introduction, line 87-88: Change to "...is a similar statistic, measuring the probability..."
Introduction, line 94: Provide the author name for the reference [41].
Introduction, line 95: Replace the capital letter in analysis.
Introduction, line 96: Change to "...the actions analyzed in the preceding studies ranged from..."
Introduction, line 98: Change to "The authors of a previous study found that movement..."
Introduction, line 104: Remove "probably" from the sentence.
Materials and Methods, lines 110-118: Provide the number of players in different playing positions (forwards, backs).
Materials and Methods, lines 121-132: Was the order of the "tackling" and "tackled" players randomized/counterbalanced? I think that this is an important element to include because you have taken the recordings from the different devices separately and presented the devices in a random/counterbalanced order; but it is important that you present as much information about the procedures as possible to show that the results were not due to an order effect. For example, let the reader know what instructions were provided to the players during each drill to ensure that the intensity of the tackles was consistent across the trials performed.
Materials and Methods, lines 141: You refer to "planes" here when earlier it was "dimensions". Please be consistent with your terminology.
Materials and Methods, line 151: Remove the unnecessary period (full-stop).
Materials and Methods, line 159: Change to "Entropy was calculated according to..."
Materials and Methods, lines 171-181: You need to provide a summary of how the standardized mean bias and the standardized typical error of estimation were calculated for the different devices. This is an important omission from the manuscript in its current form that needs to be addressed in our revisions. Presumably the bias was calculated using a method similar to the Bland-Altman approach? Is the term 'standardized typical error of estimation' correct here? You have used linear regression models and present the values for the standard error of estimation in Table 1. Is this what you mean by 'standardized typical error of estimation'?
Materials and Methods, line 178: Change to "...samples t-test was also...:
Materials and Methods, line 180: Change to "...regression analysis was performed using..."
Materials and Methods, line 181: Change to "The alpha was set a p < 0.05 for all analyses."
Results, Figures 4A, B, and C: Update the axis titles to reflect the variables being presented in each figure (i.e. Mean acceleration from 1000 Hz device). Also, acceleration presented in Figure 4A is measured in values of g rather than arbitrary units. Finally, correct the spelling in the figure titles to "linear regression".
Discussion, line 224-225: Change to "...100 Hz vs. 1000 Hz) during tackling tasks performed by male adolescent rugby league players."
Discussion, line 226: Provide space between "values" and "in".
Discussion, line 227: Change to "...to the data, a sampling frequency..."
Discussion, lines 230-233: The sentence beginning "Nevertheless..." is confusing. Prior to this you note the differences between the devices, favoring the 1000 Hz sampling frequency, and then you finish the paragraph stating that potentially the use of the high and low sampling frequencies might shed light on the tackling behaviors. Please rewrite this to ensure that your intended meaning is clear.
Discussion, line 244: Change to "Studies investigating the influence..."
Discussion, line 246: Change to "Raffalt et al. [48] have published a study..."
Discussion, line 260: Where has the value of 150 Hz come from? Neither of the preceding studies by Raffalt et al. used a sampling frequency of 150 Hz; neither did you.
Discussion, line 267: Do not use contractions.
Discussion, line 276: Change to "...load analyses."
Discussion, line 287: Change "strength exercises" to "training exercises". Undoubtedly tackling involves considerable muscular strength, but I would not immediately think of tackling drills as typical "strength exercises".
Conclusion, lines 294-296: I do not think that future recommendations should be presented in a conclusion. Simply conclude the findings of your study.
Author Response
Dear Reviewer,
Thank you very much for your comments and contributions. We believe they have significantly improved our manuscript. We have made an effort to address all your observations point by point. We look forward to hearing your feedback. You can find our responses in the attached document.
Best regards,

Reviewer 2 Report
Comments and Suggestions for Authors
1- What specific contexts or types of movement benefit most from 1000 Hz? Could more granularity on when lower frequencies are insufficient strengthen this conclusion?
Introduction
1- While the authors note that lower-frequency devices are commonly used for general movement analysis in team sports, they could expand on the unique limitations of these frequencies for accurately capturing micro-movements in high-intensity actions. Specific studies on frequency limitations and inconsistencies across contact sports would provide a stronger rationale.
2- The rationale behind using both measures (ApEn and SampEn) should be clearer, especially since they often yield similar insights. A brief comparison of why both are necessary here—such as differing sensitivities to data regularity—could enhance comprehension.
3- Given that higher frequencies capture minute changes in acceleration, it may be beneficial to explain how these changes reflect neuromuscular response differences. How do fine-grained measurements improve our understanding of physiological demands during tackles?
Methods
1- The tackle protocol is well-defined but could benefit from additional information on how closely this drill simulates in-game tackling. The controlled setup may not capture the spontaneous variability of real-match tackles. The authors could elaborate on this limitation or describe any supplementary methods to approximate game conditions.
2- : Were sessions conducted in a controlled environment (e.g., indoor training facility), or was this tested on the field? This context could affect both signal noise and the consistency of acceleration measurements, especially at higher sampling frequencies.
3- The selection of 100 Hz and 1000 Hz as sampling frequencies is well-justified, but an explanation of potential intermediary frequencies (such as 500 Hz) would be insightful. Including why these two specific rates were chosen and why they didn’t test intermediary rates would better support the rationale.
Data Analysis
1- The article could benefit from a more comprehensive explanation of error and bias interpretation for practitioners. What does a large estimation error imply for practical use? How would such error levels translate to potential in-field misinterpretations by coaches or sports scientists?
2- While large correlations indicate that general trends align across frequencies, the interpretation is somewhat limited without a discussion of residuals. Highlighting which conditions yielded the most significant deviations would clarify potential constraints for practical use.
3- The description of entropy computation (based on prior work by Goldberger et al.) could discuss how sensitive entropy measures are to sampling frequency changes, particularly with explosive sports movements. A clearer exposition of ApEn and SampEn’s ability to capture chaotic, high-variance events could enhance the depth here.
Results
1- While the higher mean accelerations at 1000 Hz make intuitive sense, the authors could expand on potential underlying reasons beyond simply better capturing peaks. Could the increased measurement sensitivity at 1000 Hz, for example, be picking up subtle nuances in collision impact?
2- Higher entropy at 100 Hz is interpreted as indicative of more chaotic signals, which might result from under-sampling key data points in rapid movements. Further detail on how this affects the practical usability of data at 100 Hz for decision-making purposes would be beneficial. Could these findings imply that standard lower-frequency sensors might overestimate variability in real-world settings?
3- The figures (particularly Figures 2 and 3) are helpful for visual comparison, but additional annotations or highlighted segments could clarify the specific time windows where frequency-based discrepancies in acceleration and entropy values are most pronounced.
Discussion
1- The authors conclude that 1000 Hz is likely more suitable for capturing high-intensity movements, while 100 Hz may suffice for general monitoring. However, more explicit frequency guidelines based on action type (e.g., collision vs. running) could be beneficial. Would 250 Hz or 500 Hz provide a reasonable balance for mixed-movement sports? Clarifying would aid practitioners in adapting these findings.
2- The authors touch on practical relevance but could go further in discussing how frequency-based data discrepancies might affect coaching decisions, load management, and athlete monitoring. For instance, could lower-frequency devices unintentionally suggest higher movement variability, leading to misinterpretations in player fatigue or collision impact intensity?
3- Addressing how well the controlled tackling drill translates to real-game scenarios would clarify the study’s applicability. Since variability in live games can be more unpredictable, future research might consider in-game data collection or mixed protocols to better reflect competitive conditions.
4- The authors mention potential improvements for sensor manufacturers but could further elaborate on practical developments. For instance, should manufacturers aim to create flexible sensors that adapt sampling frequencies based on movement type? Practical recommendations could encourage industry advancement.
5- Including a “Practical Recommendations” section in the discussion could help practitioners apply the findings, especially those without a deep statistical background.

Author Response

(The authors gave the same response as above.)

Reviewer 3 Report
Comments and Suggestions for Authors
See attached document

Author Response

(The authors gave the same response as above.)
